Variation and trade-offs in life history traits of the protist parasite Monocystis perplexa (Apicomplexa) in its earthworm host Amynthas agrestis

Keller Erin L. 1
Schall Jos. J. jschall@uvm.edu 2
1 School of Biological Sciences, Washington State University , Pullman , WA , United States of America
2 Department of Biology, University of Vermont , Burlington , VT , United States of America
Gottdenker Nicole
Electronic publication date: 2024 Mar 26
Publication date: 2024
Volume: 12
Electronic Location ID: e17161
Received 2023 Oct 8; Accepted 2024 Mar 5
Copyright: ©2024 Keller and Schall
Copyright year: 2024
Copyright holder: Keller and Schall
License: This is an open access article distributed under the terms of the Creative Commons Attribution License, which permits unrestricted use, distribution, reproduction and adaptation in any medium and for any purpose provided that it is properly attributed. For attribution, the original author(s), title, publication source (PeerJ) and either DOI or URL of the article must be cited.
License URL: https://creativecommons.org/licenses/by/4.0/

Keywords: Ecology, Life History Traits, Parasitology, Protists, Gregarines, Parasite of invasive earthworms, Parasite transmission biology

Funding: University of Vermont The work was supported by a grant from the University of Vermont to Erin L. Keller. The funders had no role in study design, data collection and analysis, decision to publish, or preparation of the manuscript.

==============================
The life history of a parasite describes its partitioning of assimilated resources into growth, reproduction, and transmission effort, and its precise timing of developmental events. The life cycle, in contrast, charts the sequence of morphological stages from feeding to the transmission forms. Phenotypic plasticity in life history traits can reveal how parasites confront variable environments within hosts. Within the protist phylum Apicomplexa major clades include the malaria parasites, coccidians, and most diverse, the gregarines (with likely millions of species). Studies on life history variation of gregarines are rare. Therefore, life history traits were examined for the gregarine Monocystis perplexa in its host, the invasive earthworm Amynthas agrestis at three sites in northern Vermont, United States of America. An important value of this system is the short life-span of the hosts, with only seven months from hatching to mass mortality; we were thus able to examine life history variation during the entire life cycle of both host and parasite. Earthworms were collected (N = 968 over 33 sample periods during one host season), then parasites of all life stages were counted, and sexual and transmission stages measured, for each earthworm. All traits varied substantially among individual earthworm hosts and across the sites. Across sites, timing of first appearance of infected earthworms, date when transmission stage (oocysts packed within gametocysts) appeared, date when number of both feeding (trophic) cells and gametocysts were at maximum, and date when 100% of earthworms were infected differed from 2–8 weeks, surprising variation for a short season available for parasite development. The maximal size of mating cells varied among hosts and across sites and this is reflected in the number of oocysts produced by the gametocyst. A negative trade-off was observed for the number of oocysts and their size. Several patterns were striking: (1) Prevalence reached 100% at all sites by mid season, only one to three weeks after parasites first appeared in the earthworms. (2) The number of parasites per host was large, reaching 300 × 103 cells in some hosts, and such high numbers were present even when parasites first appeared in the host. (3) At one site, few infected earthworms produced any oocysts. (4) The transmission rate to reach such high density of parasites in hosts needed to be very high for a microbe, from >0.33% to >34.3% across the three sites. Monocystis was one of the first protist parasites to have its life cycle described (early 19th century), but these results suggest the long-accepted life cycle of Monocystis could be incomplete, such that the parasites may be transmitted vertically (within the earthworm’s eggs) as well as horizontally (leading to 100% prevalence) and merogony (asexual replication) could be present, not recognized for Monocystis, leading to high parasitemia even very early in the host’s season.

Introduction

An organism’s life history reflects its allocation of assimilated resources into growth, maintenance, and reproduction, as well as its schedule of developmental and reproductive events over time (Roff, 1992). Trade-offs are expected across life history traits because resources allocated to one trait must subtract from others, such as the number of offspring vs. their size, and, for parasites, the effort put into evading the host immune system vs. other processes (McDade, 2003). Also limiting are developmental and resource constraints, and developmental links across traits (Eisen & Schall, 2000). These complex events play a dominant role in an individual’s fitness, so the study of life histories has long had a central place in evolutionary ecology. Important reviews include Roff (1992), Stearns (1992), and Charnov (1993); these three works have been cited an impressive ∼25,000 times.

The great majority of life history studies have centered on large multicellular free-living animals and plants, but the evolutionary and ecological theory of life histories developed for these species must also apply to micro-organisms such as protist parasites and even viruses (Neal & Schall, 2014; Wasik et al., 2014). A distinction should be made between a parasite’s life cycle and its life history. A parasite life cycle is the sequence of developmental stages, with changing morphology and physiology as the parasite enters a host, feeds, and produces the transmission stages (this may be repeated for a life cycle with two or more host species), whereas the life history is the partitioning of resources and developmental timing. How long a parasite remains in its feeding stage before switching to its transmission stage is a life history developmental decision overlain on its life cycle, as is the packaging of resources that determine the size vs. number of transmission stages. Life cycles of parasites tend to be fixed for a particular species; life history traits in contrast are notable for their plasticity. For protist parasites, lacking tissues and organs, phenotypic plasticity in life history decisions would allow a parasite to meet challenges such as changing host environments. Evolutionary theory, for example, predicts that when the host environment deteriorates, the shift to transmission stages would be accelerated (Schneider et al., 2018; Schneider & Reece, 2021). Understanding the factors that drive phenotypic plasticity in life history decisions for protist parasites, and how this plasticity influences transmission success, is still poorly understood, but should be viewed as fundamental to understanding their ecology (Reece, Ramiro & Nussey, 2009; Mideo & Reece, 2012).

Excellent systems for studies on the ecology of protist parasites are the gregarines (phylum Apicomplexa). Gregarines exploit hosts in every invertebrate group in all terrestrial and aquatic environments and display a great variety of cell morphologies, transmission tactic, and other modifications to their life cycles (Valigurová, 2012; Valigurová et al., 2023; Desportes & Schrével, 2013; Rueckert, Betts & Tsaousis, 2019). More than 1600 gregarine species have been described, but this must be a very small fraction of the actual number of taxa that exist (Rueckert, Villette & Leander, 2011). Examples of this hidden diversity abound: every insect species appears associated with one or more host-specialist gregarine symbionts (this relationship could be harmful, neutral, or even helpful Rueckert, Betts & Tsaousis, 2019; Arcila & Meunier, 2020). Environmental DNA studies of tropical soils identify a “hyperdiverse” protist biota, with a majority being gregarines (Mahé et al., 2017; Lentendu et al., 2018), and oocysts of gregarines appear common in deep-sea vent environments (Moreira & López-Garcia, 2003) and even in pelagic waters (Boisard, 2021). Gregarine DNA has been found in unlikely locations such as a swipe from a 15th century stained glass window, most likely left there by passing insects or other invertebrates (Carmona et al., 2006). Thus, there must be millions of species of gregarine parasites in all invertebrate groups in every habitat (Rueckert, Betts & Tsaousis, 2019). Plasticity in life histories of gregarines may well have allowed them to exploit such a great range in hosts in so many habitats.

Here we focus on the life history of Monocystis perplexa (Monocystidae) a gregarine parasite of an important invasive earthworm in North America, Amynthas agrestis Megascolecidae (Keller & Schall, 2020; Nouri-Aiin et al., 2022). The genus Monocystis was one of the first protist parasite taxa to have species described and their compete life cycle presented (Henle, 1845; Kölliker, 1848; Stein, 1848; Schmidt, 1854). Since then, ∼95 species have been described (Keller & Schall, 2020). Because Monocystis are ubiquitous in earthworms world-wide, they have long been featured in parasitology courses and textbooks as a classic example for the gregarines (Grove, 1923; Sheridan, 1986; Schmidt et al., 2009). However, after 175 years of studies on Monocystis (and indeed for any species in the family Monocystidae), knowledge of life history variation is absent. We recognize that a description of variation in life history traits of a parasite can only be an entry into any understanding of both ecological events and the significance of such phenotypic plasticity, but an important entry.

The earthworm host of M. perplexa follows an annual life cycle in northern Vermont, USA, hatching in early spring from eggs deposited the prior season, which are protected by a tough cocoon, then reaching reproductive maturity by July, and all dying at the first freeze in November (Görres et al., 2018; Nouri-Aiin & Görres, 2019). This aids research on M. perplexa because the complete life cycle of the parasite can be followed over those months. A. agrestis earthworms are often abundant in the upper soil levels, which leads to serious damage to soil biotic communities and disruption of soil chemical and physical properties (Görres, Bellitürk & Melnichuk, 2016), but this abundance is also convenient for collecting the earthworms to monitor the life history of M. perplexa.

Our goals in this study were: (1) To describe variation in the parasite’s life history—timing of events and how resources were partitioned—within individual earthworms, among earthworms at a site, and across sites. The sites differed in the environment offered to the earthworms (soil type, duff available for feeding, possible temperature regimes), and we expected these differences to cascade into effects on the parasite. (2) We sought correlates of this variation that could lead to understanding both their causes and effects. Our study revealed substantial variation in important life history traits within and among hosts and across sites as well as trade-offs in reproductive effort, including production of the transmission stage. (3) We found aspects of the life history of this parasite to be perplexing, and thus we present explanations that cast doubt on the classic view of the life cycle and transmission biology of Monocystis parasites. This study on life history variation thus presents a new light on the life cycle of the parasite, a life cycle thought to be well understood for more than a century.

Life Cycle of Monocystis Perplexa

To understand the cell types counted and terminology (below), it is first necessary to describe the parasite’s life cycle and developmental stages. The parasite’s life cycle is typical for Monocystis; each life stage is named and shown in Fig. 1 (see also Keller & Schall, 2020). All life stages are haploid except for an ephemeral zygote during the sexual stage. The parasite is first observed as a spherical feeding stage, or trophozoite (Fig. 1A) in the host’s seminal vesicles, the organ for storage of self-sperm. This organ is the location where a Monocystid parasite feeds and produces the transmission stages (Clopton, 2000; Keller & Schall, 2020; Schall, 2021). The trophozoites feed on the earthworm’s sperm morula structures, eventually being covered with the flagella of consumed sperm. The trophozoites then develop into the gamonts (Fig. 1B), which are flattened cells that continue to feed and grow and finally locate a mate for initiation of the sexual cycle. Two gamonts align along their long axis, a process called association (Fig. 1C), then produce a covering for syzygy when the gamonts develop into gametocytes that rotate and produce isogametes (thus, sex without gender) (Fig. 1D). Gametes fuse and develop into zygotes (Fig. 1E), and these continue development into oocysts (Fig. 1G) that fill the spherical gametocyst (Figs. 1H and 1I). Meiosis takes place within the oocyst to produce eight haploid sporozoites (Crespi, Martinucci & Ferragosti, 1981), the final infectious parasite cell. The gametocysts and oocysts are presumably liberated into the soil when the earthworm dies. We reach this conclusion because we did not find the gametocysts or oocysts in the body cavity, gut, or frass of the earthworms. Likewise, we never observed open oocysts within any of hundreds of seminal vesicles we examined, and so conclude autoinfection does not occur in this parasite’s life cycle. The next season, earthworms would then consume the oocysts from soil when feeding, and the sporozoites emerge, or dehisce, to initiate the cycle once again. Thus, Monocystis parasites, typical for most gregarines, do not undergo asexual replication in the host (“merogony” in Apicomplexa), distinct from reproduction in other clades in the phylum such as the malaria parasites that reproduce asexually in both the vertebrate and insect hosts. Monocystis replication only occurs by producing multiple gametes for each mated pair of gametocytes.

Figure 1 Life cycle stages of the gregarine Monocystis perplexa.

Life cycle stages of the gregarine Monocystis perplexa in its earthworm host. (A) Trophozoite feeding on host sperm morula; (B) mature gamont; (C) two gametocytes in association; (D) gametocytes in syzygy producing gametes; (E) Gametocyst full of zygotes; (F) gametocyst full of immature oocysts; (G) mature oocysts; (H) gametocysts packing host seminal vesicle; (I) gametocyst packed with mature oocysts.

In summary, the Monocystis life cycle stages are: feeding trophozoites and gamonts, two gamonts paired in association, gamonts develop into gametocytes, these gametocytes produce gametes, then zyogotes, then oocysts that pack the final spherical gametocyst. Gametocysts are deposited in the soil when the earthworm dies. Oocysts overwinter in the soil to be eaten by the next generation of earthworms. Asexual reproduction in the host (merogony) does not occur.

Methods

Site locations

Portions of this text (methods, data gathered, and some analysis) were previously presented as part of a theses (available at: https://scholarworks.uvm.edu/cgi/viewcontent.cgi?article=1167&context=hcoltheses). Three study sites in Chittenden County, Vermont, USA were chosen that varied in elevation and soil type; permission to sample earthworms was granted for each site by the responsible authorities. The University of Vermont Horticulture Research and Education Center (HF) is a 39 ha experimental agricultural site (76 m elevation, 44°25′53″N, 73°11′57″W), with a deciduous forest canopy of primarily sugar maple (Acer saccharum) and a soil of Windsor sandy loam. The University of Vermont Centennial Woods Natural Area (CW) is a 28 ha forest containing hardwoods, conifers, wetlands, and streams (80 m elevation, 44°28′32″N, 73°11′13″W). Earthworms were collected in Windsor sandy loam soil along a frequently used recreational path in a forest containing both hardwood and coniferous trees. Conifer needles are a major component of the soil duff at CW. The Green Mountain Audubon Center (AU) is a 100 ha educational and conservation center in Huntington, Vermont (190 m elevation, 44°20′48″N, 72°59′46″W). The area is a hilly habitat with mixed forests, fields, and wetlands and a gravel road bisecting the area. Earthworms were collected around the headquarters area in deciduous leaf debris (primarily maple and oak) and gravel. That site is the type locality for M. perplexa (Keller & Schall, 2020). The sites were 5.4 km (CW to HF) to 19.0 km (HF to AU) and 20.9 km (CW to AU) distant. The sites differ in average snowfall per year at 57.5 cm for CW and HF and 130.0 cm at AU. In summary, the three sites differed in elevation, snowfall, and the kind of leaf litter where the earthworms live and feed.

Host collection and species identification

Collection of Amynthas agrestis earthworms was done at the sites during the warm season each year from 2013 to 2022. Earthworms were identified both by morphology (Chang, Snyder & Szlávecz, 2016) and sequencing of the barcoding COI gene (Nouri-Aiin et al., 2021) to insure the same earthworm species was found at each site. Spot checks for prevalence and density of parasites were made each year except for 2015, when samples that year were taken weekly from April 5 to November 15 at each site. Earthworms were collected by manually sifting through the top 10 cm of the leaf debris and soil (O and A horizons). Sampling of earthworms took place over approximately 5 m2 at each site. After collection, earthworms were stored at 16 °C in containers with soil and leaf litter from their site of origin.

Earthworm host phenology, earthworm growth, and seasonal soil temperature

As noted above, elevation of HF and CW is similar, but higher by ∼100 m at AU. Several measures of possible influence of this topography on the earthworms were taken. (1) Differences in lifespan of the host in this annual earthworm would also reflect on the time available for the parasite’s development and production of transmission stages. Therefore, earthworm phenology was determined at each site in 2015 as date of first appearance of hatchlings, first mature earthworms (presence of the clitellum—reproductive tissue forming the cocoon that encloses an egg), and last observation of live earthworms. (2) Growth of the earthworms at each site was scored as the length of each earthworm in mm taken from week 20 (when the earthworms were sexually mature) to end of the season. (3) Soil temperature was monitored at AU and HF with a button thermal logger (Thermochron iButtons; Maxim Integrated, San Jose, CA, USA) six times a day from September 2014 to November 2016 with loggers placed seven cm below the casting layer. Several analyses for 2015 and 2016 were done to determine if there was a difference in host season length between AU and HF. First, earthworm season length was estimated by determining the longest period of consecutive days above 0 °C. Hard frosts occur at temperatures below 0 °C and results in high earthworm mortality (Görres, Melnichuk & Bellitürk, 2014); thus, consecutive days above 0 °C represents survivable conditions for A. agrestis. Second, cumulative growing degree days (GDD) was also calculated for 2015 and 2016 to determine differences in potential growth and development of A. agrestis at AU and HF. A base temperature of 5 oC was chosen because earthworm growth begins at this temperature (Görres, Bellitürk & Melnichuk, 2016). Cumulative GDD was calculated as: GDD5 = Σi [(Tmaxi + Tmini /2) −5] where Tmaxi and Tmini are the maximum and minimum temperatures on Julian date i. The accumulation rate of GDD5, representing the rate at which energy accumulates in the soil, was calculated by finding the slope of GDD5 between the first GDD5 >0 and the first GDD5 ≥ 1000. An endpoint of 1,000 GDD5 was used because development of A. agrestis adults occurs at 1,000 GDD5 after hatchlings appear (Görres, Bellitürk & Melnichuk, 2016).

Dissection of A. agrestis

During 2015, ten A. agrestis earthworms from each site were dissected within one to three days for 33 weekly samples for a total of 968 earthworms (rarely nine worms were taken for a sample). Earthworms were killed by submersion in 50% ethanol, blot dried, and examined under a dissecting microscope. Earthworm segments 1–20 were bisected on the ventral side and, if present, the seminal vesicles removed with forceps and placed into a finger bowl. Seminal vesicles are soft tissue and thus readily manually homogenized with forceps in a known volume of earthworm saline solution (Schall, Nouri-Aiin & Görres, 2022). For counts, 1 µl of the homogenized sample was used to make a live microscope slide preparation and observed under a cover slip. The volume of the seminal vesicles was found by subtracting the volume of the saline solution from the total volume determined after counts were made by drawing the mixture into a 1 ml syringe.

Parasite identification, phenology, and density in hosts

Identification of M. perplexa was based on morphology (Keller & Schall, 2020) and sequencing of the 18S rRNA gene (Keller & Schall, 2020). The sequences were identical for parasites from all three sites. Presence of M. perplexa infection was determined by microscopic scanning the preparation for parasites at 400x. If no parasites were observed in 1 µl, up to 5 µl of the sample was scanned for parasites, and earthworms with no parasites present in 5 µl were considered not to be infected in the seminal vesicles. For infected samples, M. perplexa parasite stages were scored as trophozoites, gamonts, gametocytes, gametocysts with developing oocysts, and gametocysts packed with mature oocysts (Fig. 1). Total parasitemia for each life stage seen in the earthworm’s seminal vesicles was determined by multiplying the number of parasites observed per unit volume by the total volume of the seminal vesicle. The totals were then grouped into “trophic” parasites which is the sum of trophozoites, gamonts, and gametocytes, and “gametocysts” containing either developing or mature oocysts. Hence, the terms used here will be “trophic stages” and “gametocysts”.

Comparison of reproductive output at two sites, AU and HF

For some infected earthworms in 2015, when the presence of mature gametocysts was detected, gentle pressure was applied to the coverslip of the microscope slide to compress the oocysts within each gametocyst into a single layer. Photographs of a compressed gametocyst were taken through a Nikon microscope at 400x using an attached Moticam 1,000 1.3MP Live Resolution (Richmond, British Columbia) microscope camera and Motic Image Plus 2.0.11 program. Thirty gametocysts were photographed for ten infections each at AU and HF for a total of 600 gametocysts and counts of the number of oocysts per gametocyst were taken. Gametocysts were rarely observed in CW infections over the course of the 33 weeks (below); thus counts from some CW gametocysts were done, but were excluded from the site comparison of oocyst production. Also, to investigate the relationship between oocyst size and number of oocysts per gametocyst, length and width measurements of five oocysts per gametocyst were taken from 30 gametocysts used to determine the reproductive output at HF.

Measurement of resource acquisition by parasites before sexual reproduction was calculated by measuring the area of the gametocytes (Fig. 1C). During the week of 23 July, 2017, gametocytes were measured at AU (N = 60 from four infections) and HF (N = 44 from four infections). The length and width of the gametocytes were measured at 400x using the Moticam system. Last, the cross-sectional area of the fusiform oocysts was also measured (using length and width) to determine the relationship between number of oocysts packed within a gametocyst and the size of 5 oocysts from 30 gametocysts which had the oocysts counted (total = 150 oocysts). Because oocysts change their morphology during development, only mature oocysts were extracted from the gametocysts for measurement.

Transmission success of oocysts

The parasite’s transmission success is the proportion of the final transmission stage (sporozoites within the oocysts) that survive to produce parasite cells within the next year’s generation of hosts. We calculated this measure by assuming the total number of parasite cells observed during the focal year would be equal the following season. For the focal year, the total number of parasite cells was the sum of all cells (trophozoites, gamonts, gametocytes, and 2 ×the gametocysts), and the total number of transmission stages was calculated as the total oocysts ×8 (because each oocyst carries eight sporozoites that infect an earthworm (Crespi, Martinucci & Ferragosti, 1981). The transmission rated needed to maintain the parasite population from year-to-year would be the total parasite cells divided by total sporozoites. That is, the calculation estimates the percent of the sporozoites produced during the focal season that would need to survive to yield a stable number of parasites the following season.

Calculations and analysis Analysis was performed on the JMP Pro platform (SAS Institute, Cary, NC).

Results

Soil temperature, earthworm phenology and growth

During the period of study, snow/ice remained at soil surface each year ∼7 days longer at AU than the other sites which is expected given the higher snowfall at that site. This is reflected by the date in 2015 when soil temperature rose above 0 °C (April 14 at HF and May 3 at AU). Overall though, soil temperature tracked closely at AU and HF where thermal loggers were placed (Fig. 2). However, the soil at AU was warmer over the entire season, with GDD 2,130 at AU and 2,048 at HF, with the slope from 0 to 1,000 GDD of 11.9/d at AU and 9.6/d at HF. Number of consecutive days when soil was >0 °C was also longer at AU than HF (262 vs. 252). Thus, although AU was at higher elevation and retained snow/ice on the soil surface longer, overall the soil at AU was warmer and the opportunity for the earthworms to grow was longer.

Figure 2 Soil temperature recorded at two sites in the study.

Mean weekly soil temperature recorded at two sites in the study, AU and HF, over a two year period. Temperature profiles were very similar for the two sites. Hatchling earthworm hosts were first seen in April; note this is soon after soil temperature rose above 0 °C. All earthworms died after first hard freeze in November.

Hatchling earthworms at each site were observed by sampling under the snow/ice in early April, and all earthworms died after the first hard freeze at each site (week 33 for 2015). However, size of earthworms in mm total length (N = 135 –140) from week 20 to end of the season was very similar across sites: x (SD) = CW 99.2 (19.5), AU 100.5 (18.6), HF 98.6 (16.0); ANOVA F = 0.3927, P = 0.675.

Parasite phenology

Summary of parasite phenology and infection characteristics are given in Table 1, so values for the variables can be consulted there. Figure 3 presents the median and range each week for trophic cells and Fig. 4 gives those results for gametocysts. The earthworms did not develop seminal vesicles (the site of infection) until July, and first observation of parasites was at week 16 for CW, followed 1–2 weeks later at the other sites; thus no infected earthworms were observed from samples taken April 5 to July 12. The parasites, therefore, had only ∼3.5 months to develop and produce transmission stages before all earthworms died in November. Trophozoites were the first stage seen at each site, followed by gamonts, gametocytes, and finally gametocysts. This is expected as this is the sequence of development in Monocystis spp. (Fig. 1). Prevalence reached 100% at all sites, just 1 week after parasites were first observed at CW and 3 weeks for the other sites.

Table 1 Parasite phenology and infection characteristics at three sites for the gregarine Monocystis perplexa in its earthworm host Amynthas agrestis.

Earthworm hatchlings were first observed at Week 1 (early April), through midseason (Week 17 in mid July), and to final observation of earthworms before they all died in mid November (Week 33). Trophic stages = parasite cells other than gametocysts; Minimum oocysts eaten per host = the number of oocysts that would yield the total number of parasite cells. Few infections produced gametocysts at CW, so data on gametocyst parasitemia not given. Further description of variables given in the text.

	Audubon Center (AU)	Horticultural Center (HF)	Centennial Woods (CW)	
N Earthworms	161	151	179	
Week First Infected	17	18	16	
Week 100% Prevalence	20	21	17	
Week Peak Parasitemia	
trophic stages (median number of cells)	28 (219.9 × 103)	25 (218 × 103)	20 (92 × 103)	
Maximum parasitemia trophic stages	658.2 × 103	328.3 × 103	174.2 × 103	
Week first gametocysts	17	21	18	
Week peak parasitemia gametocysts (median number)	28 75.1 × 103	29 25 ×  103		
Maximum parasitemia gametocysts	227.2 × 103	69.8 × 103	6.9 × 103	
% hosts with gametocysts	78	90	6	
Minimum oocysts eaten per host. Mean (SD), range	16,662 (13,244)
75–98,552
N = 145	12,036
(7,655)
27–43,374
N = 122	4,720
(5,139)
15–21,777
N = 156	

Figure 3 Density of trophic stages of Monocystis perplexa cells in host seminal vesicles.

Trophic stages are shown in (Figs. 1A–1D) over time (weekly counts). No parasites were seen from Week 1–15 (early April to mid July). Median value shown by horizontal bar and range by vertical line. Three sites used in study are shown.

Figure 4 Number of gametocyst stages of Monocystis perplexa in host seminal vesicles.

Gametocysts are shown in Figs. 1E–1F, 1H, 1I over time (weekly counts). Median shown by horizontal bar and range by vertical line. Two sites are shown.

Peak parasitemia of trophic forms was scored as the week when their median parasitemia was greatest, and was reached at CW 5–8 weeks earlier than the other sites. After peak parasitemia, the density of trophic cells began to decline when gametocysts were produced (Figs. 3 and 4). That is, trophic cell counts declined when they made the transition into gametocysts. At least some trophic cells must make this transition very quickly, entering into the sexual cycle, because gametocysts were observed the same week as first parasites in two earthworms at AU, and only 2–3 weeks later at the other sites. Another reflection of a rapid development of gametocysts was the low numbers of cells in association or syzygy observed compared to gametocysts. Thus, the association and syzygy stages must be ephemeral, and develop quickly into gametocysts.

Parasite density in hosts

A striking observation was the high density of parasites observed in some earthworms as soon as infections were observed (Figs. 3 and 4). For example, at the week parasites were first observed at AU, one earthworm carried 240 ×103 trophic parasite cells. That individual must have eaten 30 ×103 oocysts (assuming full success of the eight infective cells per oocyst) since hatching, or ∼175 oocysts per day. For the week when peak parasitemia of trophic cells was observed, the median number of cells ranged from 92–219 ×103 across sites. Gametocyst density in the earthworms later in the season was high, with gametocysts often packing the seminal vesicles.

A calculation was made of the minimum number of oocysts that each earthworm must have eaten to yield its observed parasite density (N trophic cells + 2 ×N gametocysts), and shows the earthworms had eaten as many as 98 ×103oocysts (Table 1). Again, there was a site effect, with the maximum number of oocysts eaten differing greatly from the highest at AU, then HF, and finally CW (ANOVA, F = 70.9, P <<0.001, with posthoc tests showing each pair of sites with P < 0.001). The actual number, though, must have been much higher because not all oocysts eaten would have produced a full complement of successful infective cells. For example, if viability of the sporozoite cells contained within oocysts was 50%, some earthworms must have eaten 2 ×105 oocysts.

Although parasite phenology was more rapid at CW than other sites (that is, earlier when parasites first seen, 100% prevalence, peak parasitemia, and first production of gametocysts), the peak parasitemia of trophic parasites at CW was only half that of the other sites. More perplexing was the low production of gametocysts at CW. These contain the transmission stages, yet only 6% of infections at CW produced any gametocysts, compared to a very high proportion of infections at the other sites producing gametocysts. The parasitemia of gametocysts at CW was also much lower than at the other sites (10–32x greater at the other sites). Thus, a trade-off was seen, with rapid events associated with lower production of parasites.

Measures of transmission

Production of oocysts depends on resources assimilated by the developing trophic-stage parasites. Therefore the size of gametocytes in association (when the cells have ceased feeding, Fig. 1C) was measured and compared between AU and HF (few final gametocytes are produced at CW, so that site was not included). HF gametocytes were larger in size than those at AU (Fig. 5A, HF x = 1016 µm2, AU x = 675 µm2; Nested ANOVA with parasites nested within earthworms and earthworms nested within sites, P < 0.001). Individual earthworms accounted for 46.5% of the variation. Thus, parasites at HF processed more resources into these cells, although ∼ half of the variation was due to effect of individual host. This larger assimilation of resources into gametocytes at HF should result in greater production of oocysts: this was true (Fig. 5B). The number of oocysts per gametocyst, was compared using a nested ANOVA with counts for oocysts nested within worm and worms nested within sites. The number of oocysts per gametocyst differed among earthworms at a site (nested ANOVA, P < 0.0001) and between the two sites (P = 0.0027), with 97% of variation attributed to individual infected earthworms. The mean number of oocysts per gametocyst was higher by 3% at HF (145) than AU (140), thus yielding 40 more sporozoites produced per gametocyst at HF (Fig. 5B). Another indication of how resources are partitioned by the parasite cells comes from a negative relationship between oocyst size and number of oocysts per gametocyst (Fig. 5C).

Figure 5 Production of transmission stages, the oocysts, based on assimilation of resources into gametocytes and then the oocysts.

(A) Area of gametocyst stages in association (Fig. 1C) at two sites, AU (N = 60) and HF (N = 44). Horizontal bar = median, box = interquartile range, whiskers = range of data. Several cells shown as circles were > 1.5 first quartile or < 1.5 third quartile. (B) Counts of oocysts per gametocyst for two sites, AU and HF. (C) Trade-off between number of oocysts per gametocyst and their size. Each point represents the mean for five oocysts counted for each of 30 gametocysts.

In summary, the parasites varied in their investment into gametocysts among earthworms and across sites, with larger cells produced at HF, and this is reflected by a larger number of oocysts produced at HF. However, twice the variation was accounted for by individual hosts for oocyst production compared to gametocyte size, and gametocytes were 50% larger at HF, but oocyst production was only 5% larger. There was a trade-off between the number of oocysts produced by a gametocyst and their size. This all concludes that larger gametocytes yield more oocysts, but smaller oocysts. Again, the smaller oocysts may have lower viability, and thus a lower production of the oocysts may yield larger ones that are more likely to survive.

A striking difference across sites was seen in the production of oocysts and the number required to maintain a stable parasite population into the next generation (Table 2). The number of oocysts produced shows the total number for all infections at AU and HF were much greater than at CW (300-fold and 163-fold greater). To produce the observed prevalence at each site, the transmission success rate would thus need to be very high for a microbial parasite, and exceptionally so for CW.

Table 2 Expected transmission success of oocysts of Monocystis perplexa.

Calculation of transmission success of the gregarine parasite Monocystis perplexa required to maintain total parasite numbers from focus year to the next season. All values are in 106 cells, oocysts or sporozoites. Given are total number of parasite cells in all collected earthworms during focus year, total number of oocysts (the stage entering the soil to overwinter until eaten by hosts the next season), the number of sporozoites within those oocysts (eight produced per oocyst), and the percent of the sporozoites that must produce viable infective cells in the earthworms. Results are provided for three sites.

	Audubon Center (AU)	Horticultural Center (HF)	Centennial Woods (CW)	
Total parasite cells	17.4	12.6	6.0	
Total oocysts produced	659.3	358.2	2.2	
Total sporozoites produced	5,274	2,866	17.6	
Minimal transmission	>0.33%	>0.44%	>34.3%	

Discussion

As organisms acquire resources they make developmental decisions that are central to their fitness. These include how long to feed, how large to grow, when to switch to reproduction, and number and size of the offspring they produce. Understanding these life history events has long been a major effort in evolutionary ecology (Roff, 1992). The first goal is to understand how life history traits vary within a species and the origin of that variation. We examined a suite of life history traits of a gregarine parasite, Monocystis perplexa, in its host the invasive earthworm Amynthas agrestis, including timing events (such as when the transmission stages were produced) and resource allocation (final size of trophic cells and how many transmission cysts produced). All traits varied substantially, both among individual infected earthworms and populations across three environmentally-distinct sites. For example, the transmission-stage gametocysts were first seen fully a month earlier at the AU site than at the HF site when the season for parasite development lasted only about 3.5 months. The density of parasites in a host, even for a single sample period at a site, ranged from hundreds to hundreds of thousands. Several factors could give rise to this great variation. These are host feeding behavior, environmental conditions experienced by the earthworms and thus the parasites, intrinsic phenotypic plasticity of the parasite, and the genetic diversity of both parasite and earthworm host. We review life history variation in M. perplexa, and ask how these four factors could explain that variation. Last, we note anomalies in the results that cast a new light on the life cycle of Monocystis.

Asexual replication of the parasite within the host (merogony) is assumed absent (but see below for this fundamental assumption to be questioned), so the arrival and density of M. perplexa cells would be a function of the earthworm feeding behavior. The earthworm’s rate of ingestion of chosen foods and the precise location chosen for feeding would determine the influx of oocysts, and then the liberation of infective sporozoites as the oocysts dehisce. The density of oocysts in the soils (left there by the previous season’s earthworms) could well vary across very short distances, even just a few cm. Wherever an earthworm died that previous season, there would be a high-density patch of oocysts. Thus, earthworms could be consuming the parasite oocysts at different rates even at a single site.

Environmental factors must influence the earthworm feeding and then the viability of the oocysts and the sporozoites that travel from the gut to the seminal vesicles. For other gregarine species, host environment is known to influence the parasite’s cell morphology, survival, and transmission success. This includes temperature, with higher temperature stunting development or even eliminating the infection (MacDougall, 1942; Patil, Patil & Patil, 1983; Kolman, Clopton & Clopton, 2015), host nutrition (Schreurs & Janovy, 2008; Hoang et al., 2017), and host species for generalist gregarine species (Hussain et al., 2013). In our study, the quality of the study sites for the health of the earthworms could vary because site AU had warmer soil, albeit only slightly, and pine needles made up a major component of the duff layer fed upon by the earthworms at site CW that may not be as nutrient-rich as maple and other deciduous leaves. Pine needles also contain compounds not present in angiosperm leaves, such as high concentration of phenolics, which may explain some of the odd life history features seen at CW (below). Yet the growth of the earthworms, a measure of overall environmental quality, resulted in very similar body sizes for the three sites. Perhaps the variation we observed was driven by fine-grained differences in environmental quality (the microenvironments faced by individual earthworms). This is possible, because over the years of the study we frequently noticed that the earthworms were found in clusters of three or more individuals within a few cm in the soil. A. agrestis earthworms are parthenogenetic (yet still produce sperm and the seminal vesicles for storage of self-sperm (Keller, Görres & Schall, 2017; Nouri-Aiin et al., 2022), so those clusters of earthworms were not mating groups. Instead, earthworm clusters might occur at high-quality environmental patches. We observed that the Amynthas earthworms are very active. Their colloquial names are “jumping worms” because they can jump several cm when held, and “snake worms” because they can travel across the surface of the soil quickly in the manner of small snakes, so they could easily seek out nutrient-rich or thermally attractive patches of the environment. Even slight differences in temperature or foods available to the earthworms would cascade to effects on parasite density, growth, and development.

Intrinsic parasite effects, especially phenotypic plasticity and trade-offs, are well known for protist parasites (Birget et al., 2017; Reece, Ramiro & Nussey, 2009). The proportion of oocysts that dehisce, and the proportion of parasite cells that enter the seminal vesicles and survive would require some developmental decisions based on rate of incoming oocysts (a crowding effect) and the challenges of physiological variation in the host based in part on temperature and diet. Other traits are more likely driven by the parasite. The shift from the trophic stages to the sexual cycle and production of transmission oocysts, one of the most important developmental events for any parasite, differed by weeks across sites. Investment traits revealed substantial differences also. The proportion of infections that produced gametocysts packed with oocysts varied greatly from 90% to only 6% across sites. Surely production of the gametocysts is the central event in the life cycle of M. perplexa, so this dramatic difference across sites is perplexing. Evolutionary theory predicts a rapid switch to reproduction when the environment deteriorates, and this has been experimentally demonstrated in malaria parasites (Buckling et al., 1997). The physiological stress experienced by the earthworms at different sites, and even environmental patches, are unknown but must have consequences for the expected success of Monocystis parasites.

Trade-offs are expected across life history traits, with limits based on developmental and resource constraints, and developmental links between traits (Eisen & Schall, 2000; Roff & Fairbairn, 2007; Neal & Schall, 2014). An obvious example is the constraint of size of offspring vs. their number such that production of larger offspring is possible only by reducing their number. For M. perplexa, the size of the gametocyte after the cell has ceased feeding was correlated with the size of the gametocyst and the number of oocysts produced there. Thus, a larger assimilation of resources by the feeding cells ultimately leads to more oocysts, yet there is a negative relationship between oocyst number and size. Important missing information is the relationship between oocyst size and their durability in the soil over winter, and the viability of the sporozoites as they emerge in the next generation of hosts. Also perplexing is why some cells cease feeding early, even in the same host, when other cells continue to feed and ultimately produce more oocysts. For gregarines, this pattern exists not just within species, but among species with variation in gametocyst size and then the number of oocysts (Borengasser & Clopton, 2019).

The last possible driver underlying the observed variation in life histories would be genetic diversity, both in the earthworm host and the parasite. Genetic background is known to influence the morphology of gregarine cells (Sander et al., 2013) and survival of a gregarine’s transmission stage (Sánchez et al., 2021), but genetic diversity and geographic population structure of gregarines in general is very poorly studied. For M. perplexa, spatial (among site) genetic structure of both parasite and host may be present. Information on the genetics of M. perplexa is lacking, but for another Monocystis infecting Lumbricus terrestris in Germany multiple genotypes of parasite, often nine per host, were detected in individual earthworms (Velevan, Schulenberg & Michiels, 2010; Weller, 2013). The genetic picture of A. agrestis earthworms reveals they are parthenogenetic, reproducing clonally, yet there are multiple genetic clones at each site (Keller, Görres & Schall, 2017; Nouri-Aiin et al., 2022). If each site was seeded by a different combination of these clones after being introduced from their likely origin in Japan (Keller, Görres & Schall, 2017), the parasites may also differ genetically and face differences in the earthworm behavior or physiology. Invasive species, such as A. agrestis, are predicted to have low genetic diversity due to small propagule pressure as they entire their new habitat (Keller & Schall, 2020), and likewise the parasite would experience a similar genetic bottleneck. But, at least for the host, A. agrestis, this seems not to be the case.

In summary, the critical measure of fitness for M. perplexa is the number of transmission stages produced. Focusing on that measure, the AU site had highest density of trophic parasites per host, number of gametocysts, and final oocyst production. Site CW had the lowest production of transmission stages. This difference is not trivial, by a factor of two for site AU vs. site HF, and fully greater than two order of magnitude for AU vs. CW. For a maximum production of M. perplexa transmission cysts, a long series of events are required: the rate of acquiring the infective stages (earthworms eating the oocysts) must be high, the survival and density of the parasite cells high, the shift from trophic cells to the sexual cycle and development of gametocysts rapid, oocyst production per gametocyst high, size of oocysts large, and final gametocyst production high. CW seems the outlier, with earlier appearance of parasites and production of gametocytes, yet much lower final density of parasites and very low production of the gametocytes. Achieving all of these checks would face the challenges noted, from feeding behavior of the earthworms, to health of the cells based on host diet and environment, and any trade-offs present. A complex mix of these factors, all interacting, could well explain the differences seen among individual earthworms and the sites.

Other findings of the study remain particularly puzzling. First, fully 100% of the earthworms were found infected by mid-season at all sites. Very high prevalence of Monocystis parasites in earthworms may be the rule. For example, Field, Schirp & Michiels (2003) noted that the Monocystis infecting L. terrestris in Germany also reaches almost 100% prevalence. Second, the density of parasite cells in many hosts, even early in the season, is very high. The earthworms would need to eat many thousands of oocysts to yield such massive parasite loads. Third, at one site, CW, few infections produced the transmission stage, with most being dead-ends for the parasite, yet parasite density in hosts was very high even at that site. Last, the transmission success of the oocysts needs to be very high, even >34% at one site (CW), to maintain the high proportion of earthworms infected and the high density of parasites seen in hosts. If our counts of parasites were underestimates, then the required success would need to be even higher.

These issues would be resolved if the accepted life cycle of Monocystis is incomplete. Epidemiology theory finds that a very high prevalence of a parasite, even 100% as we found, is expected when there is both horizontal and vertical transmission (Busenberg & Cooke, 1993; Lipsitch et al., 1995). For the Monocystis system, the earthworms could become infected by ingesting the oocysts (horizontal transmission which is the accepted mode of transmission for Monocystis), but also by movement of the parasite into the earthworm’s eggs (vertical transmission). Vertical transmission of Monocystis has not be included in the accepted life cycle of the parasite, but would readily account for the very high prevalence in earthworm hosts. This might explain the 100% prevalence of M. perplexa, but not the high density of parasites in the earthworms. The earthworms would need to eat many thousands of oocysts to yield such high parasite loads, and if some earthworms become infected only by vertical transmission, the parasitized egg cocoon would have to be packed with parasite cells.

A resolution would be if merogony, or asexual replication as observed in the most other Apicomplexan parasites, is actually present. If so, the density of the parasites within the host could be driven by a classic life history trait for protists, their rate of replication. No merogony has ever been noted for Monocystis, and is considered rare in gregarines. Gregarines that do utilize merogony were once placed into a single clade (Neogregarines), but well-supported phylogenies show the Neogregarines are polyphyletic, such that merogony has evolved multiple times across the gregarines (Cavalier-Smith, 2014). Cavalier-Smith (2014) proposed that multiple fission is “easy to evolve” and suggested that merogony is more common in the gregarines than now accepted. If this is true for M. perplexa, where is that replication occurring in the host? Such cells may be difficult to identify only on morphology seen under the light microscope because they would be spherical cells very similar to some classes of the earthworm immune system cells, the coelomocytes (Stein, Avtalioin & Cooper, 1977). The origin of coelomocysts is still debated, but generally thought to derived from specialized regions of the body epithelial lining (Cooper & Stein, 1981). Merogony of M. perplexa could also take place along the body wall very early in the season, with the parasite moving en masse to the seminal vesicles. Gregarine parasites that replicate via merogony in their insect hosts are often highly virulent (Maharramov et al., 2013; Yaman, Kıran & Radek, 2023), but we observed no evidence of such harm in the host of M. perplexa.

If these suggestions are valid, that would mean our view of the life cycle of Monocystis needs to be revised to include vertical transmission and merogony that leads to very high prevalence and high density of the parasites in most hosts. As noted above, the life cycle of Monocystis was one of the first described for protist parasites, more than 150 years ago, so this suggestion would certainly be noteworthy.

Conclusions

We measured life history traits for M. perplexa based on population measures (for instance total parasites in a host) as well as measurements of individual parasite cells (including the terminal size of the cells entering the sexual cycle). All life cycle stages for the parasite ultimately lead to development of transmission-stage oocysts that determine the parasite’s fitness. Thus, selection will favor a life history that maximizes production of successful oocysts. This is the long-standing prospective of life history theory, including for parasites, that the currency of natural selection is offspring (Poulin, 1996). However, we found substantial variation in all measured life history traits, including for the parasite density in individual earthworm hosts and the number of oocysts produced by those infections. In some cases, this variation could be explained by trade-offs such as final size of the cells before initiation of the sexual cycle vs. number of oocysts produced by the mating cells. The trade-off would be the speed of production of the oocysts (by ending feeding earlier) vs. the number produced. Such trade-offs must represent alternative solutions to developmental events that yield equal reproductive success. This is another conclusion that is general in ecology, but is not really resolved (Roff & Fairbairn, 2007). Indeed, demonstrating the adaptive value of any form of phenotypic plasticity, not just for life history traits, has long been recognized as a difficult, even onerous, task (Maynard Smith, 1978).

More problematic is the great variation seen in number of parasites per host. This variation could result from the number of oocysts consumed by the earthworms (and thus out of the parasite’s control) as they feed, but also the physiological state of the host based on nutrients (and toxic compounds) in the food, temperature, and moisture levels, all of which could alter parasite development. Poulin (1996) reminds that “Parasites may well inhabit the host, but the host inhabits the external environment.” The little information we have on the feeding behavior of Amynthas is that they are generalists, taking a great variety of food types in the soil, and can readily adjust their diet depending on prevailing environmental conditions (Zhang et al., 2010), but how this would alter the physiological state of the parasite is unknown. Note that we selected sites that appeared to differ in environments offered to the hosts, but growth of the earthworms was very similar among the sites, close to identical, suggesting that they could buffer environmental variation (and may itself would partially account for the high success of these invasive soil organisms). We expected the effect of site to be dominant on the development of the parasites, but we found instead that individual earthworms determined most of the variation in life history traits.

We conclude that the very high density of parasites in individual worms, even early in the season and even at a site where few oocysts are produced by the hosts can be explained by presence of asexual parasite replication in the host, unknown until now in Monocystids. An earthworm could become infected by eating only a few oocysts or by a small inoculum carried in the egg cocoon, yet rapid asexual replication would lead to a high density of the Monocystis cells. But if this is correct, what are the triggers that yield rapid vs. slow replication, and what would control asexual reproduction to yield such variation in final density of parasites within a host? Again, if the host environments can differ and even change over the earthworm’s lifetime, the parasite may alter its rate of replication as a plastic life history response.

Most perplexing in this study is why so many infected earthworms showed little or no production of the transmission stage oocysts. This failure to produce oocysts ranged from 10% of infections at some sites and fully 94% at one site. Even for infections producing the oocysts, many parasite cells were still in the trophic forms right to the end of the lifespan of the earthworms. (This would resemble finding an annual plant species with many individuals producing no seeds, and 90% of the plants at a site lacking any seed production.) Gametocysts full of oocysts were seen in the earthworms early in the season (as expected by life history theory for a very short life span of the host), so the sexual cycle with mating of the mature gamonts can be very rapid. Is this finding unique for M. perplexa, or is it also the case for other Monocystid parasites? Again, there must be some trigger that drives a switch from the feeding trophic parasite stages toward the sexual cycle and final yield of transmission stages, but the nature of that trigger remains enigmatic. Information on the physiological factors, both in the earthworm and the parasite, that determine developmental events in M. perplexa, and indeed any Monocystid parasite is lacking, but studies are sure to open very interesting windows into the biology of gregarines. Last, data on the genetic variation of Monocystids within individual earthworms is scant. Perhaps there is mate-choice based on genotype by the parasites, and if suitable genetic matches (such as avoiding inbreeding) are lacking, there is no sexual cycle and thus no oocysts produced.

All of these puzzles can only be resolved by experimental studies on these important invasive earthworms. We note that Amynthas agrestis earthworms are readily kept in the laboratory from hatchling stages to adults (Nouri-Aiin & Görres, 2021), so observations on the effect of foods and environments presented to the hosts on the consequences for the Monocystis parasite would be very productive.

We thank Josef Görres for introducing us to the invasive Amynthas earthworms and the field sites. Maryam Nouri-Aiin assisted with field work during the later years. Charles Goodnight offered advice on the analysis.

Additional Information and Declarations

Competing Interests

Author Contributions

Data Availability

The authors declare there are no competing interests.

Erin L. Keller conceived and designed the experiments, performed the experiments, analyzed the data, authored or reviewed drafts of the article, and approved the final draft.

Jos. J. Schall conceived and designed the experiments, performed the experiments, analyzed the data, prepared figures and/or tables, authored or reviewed drafts of the article, and approved the final draft.

The following information was supplied regarding data availability:

The data is available at Zenodo: Schall, J. J. (2023). Variation and trade-offs in life history traits of the Apicomplexan gregarine Monocystis perplexa in its earthworm host Amynthas agrestis. https://doi.org/10.5281/zenodo.8332655.

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
