# Peer review of "Variation and trade-offs in life history traits of the protist parasite Monocystis perplexa (Apicomplexa) in its earthworm host Amynthas agrestis"

_PeerJ, doi:10.7717/peerj.17161_

## Round 0.1 · original submission · Major Revisions

Both reviewers find that your article studying the variation in life history traits of the apicomplexan gregarine Monocystis perplexa in an earthworm species is a well-designed and carried out study valuable to the field, but have included important comments to further clarify some findings, as well as to define and clarify aspects of statistical analysis and results, particularly in the description of identification methods, the definition of cell counts and transmission rate. I believe these critiques will not be not difficult to address. Please see reviewers comments for further details.

**Language Note:** PeerJ staff have identified that the English language needs to be improved. When you prepare your next revision, please either (i) have a colleague who is proficient in English and familiar with the subject matter review your manuscript, or (ii) contact a professional editing service to review your manuscript. PeerJ can provide language editing services - you can contact us at [email protected] for pricing (be sure to provide your manuscript number and title). – PeerJ Staff

Reviewer 1 ·

Basic reporting

The manuscript presents a study of life history traits of the gregarine Monocystis perplexa (host: the invasive earthworm Amynthas agrestis) through three different sites and over the entire life cycle of both. The authors have provided a comprehensive introduction and background, effectively situating their work within the wider landscape of the study of life history traits in parasites, particularly in apicomplexans. References are appropriate and demonstrate a thorough knowledge of existing research, highlighting the nuances and advances that their study highlights. However, some sections could benefit from further elucidation and disambiguation to improve comprehension. Some parts (indicated below and in further sections) may need rewording to be explicit and unambiguous. Resolving these issues would improve the manuscript and provide a coherent reading experience.

The manuscript has an adequate structure. All figures presented in the article are relevant to the content, facilitating visualization and understanding of the concepts discussed. Tables provide useful data summaries that support the text. They are presented in a clear format, making the data accessible and easy to interpret. The authors have provided the data on which their conclusions are based. To ensure transparency and compliance with disciplinary standards, data have been deposited on Zenodo, a reputable and recognized repository.

I have noted below the passages in the summary that would be worth revising for clarification. I have identified other passages in the body of the text that would benefit from clarification in the following sections to enhance the manuscript readability.
l. 24-26: “The life history of a parasite (resource partitioning and timing of developmental events), is overlain on its life cycle (the sequence of morphological steps as it feeds and finally produces the transmission stages).”
>>> To enhance the introduction of the abstract, I suggest refining the mentioned sentence for improved clarity. Specifically, it would be beneficial to more distinctly delineate the differences between 'life history' and 'life cycle'.
For example:
“The life history of a parasite, encompassing resource partitioning and the precise timing of developmental and behavioral events, is intricately intertwined with its life cycle, which charts the sequence of morphological stages from feeding to the transmission phases.”

For clarity stake, I would advise to rephrase lines 47-48 as follows:
“The transmission rate to reach such high density of parasites in hosts needed to be very high for a microbe at two sites (> 0.44%)”
>>>
“The transmission rate for one parasite to reach such high density of Monocystis perplexa in hosts needed to be very high at two sites (> 0.44%).”
Also, it is even higher in the third site? The formulation is confusing.

Experimental design

The authors have clearly defined their research question, which is both relevant and important to the study of gregarines. Quite rightly, there is a lack of evaluation of the life traits of these parasites. The present study applied to Monocystis perplexa will hopefully inspire other analyses of the life history traits of gregarines.
The methodology and results presented in the paper are appropriate and within the scope of the journal, offering new contributions to the body of knowledge and stimulating hypotheses for future research perspectives. The investigation seems to have been conducted rigorously and according to the field’s current standards.

Suggested improvements:

1/ The manuscript lacks a dedicated methodology section for statistics. Some of the information can be found in the results section, but it needs to be specified and detailed in the methodology section (tests, software).

2/ The count of cells is a bit hard to follow.
First it is stated that:
l. 256-258
“Total ‘immature’ parasites is presented as the sum of trophozoites, gamonts, and gametocytes (2x the number of cells in association), and ‘gametocysts’ as both the total number of immature and mature gametocysts”
Then later when describing the transmission rate calculation:
l. 286-290
“We calculated this measure if the total of number of parasite cells one year was replicated the following season: total number of immature parasite cells plus 2x the number of gametocysts divided by the total number of sporozoites produced (number of oocysts x 8, because each oocyst carries 8 of the final transmission stage; Crespi, Martinucci & Ferragosti, 1981)”
a/ First sentence is confusing: what do you mean by “replicated”?
b/ 2x the number of gametocysts: I assume it is to take into account the fact that 2 gametocytes are needed to form a gametocyst, but it would be beneficial to explain this part in a more detailed manner.
c/ Typo in “sporozoites”

3/ In the following line, please detailed the instrument used; I assumed it is the same described later on l. 265-266.
l. 249-250
“Presence of M. perplexa infection was determined by scanning the preparation for parasites at 400x.”
l. 265-266
“using a Moticam 1000 1.3MP Live Resolution (Richmond, British Columbia) microscope camera and Motic Image Plus 2.0.11 program”

Validity of the findings

The experimental design appears appropriate and well controlled. Potential flaws and blind spots are mentioned and discussed. The statistical analyses appear to be appropriate for the data evaluated.

The manuscript's conclusions align with the original research question and are grounded in the results obtained. While parts of the discussion are speculative, these sections highlight important gaps in our current understanding and suggest relevant areas for future research.

Suggested clarification and improvements:
l. 325-329
“After peak parasitemia, the density of immature cells began to decline as gametocytes were produced (Fig. 3, 4). At least some parasite cells must developed very quickly, entering into the sexual cycle and producing mature gametocysts because gametocysts were observed the same week as first parasites in 2 earthworms at AU, and 2 – 3 weeks later at the other sites.”
Shouldn’t be the first bold occurrence be gametocysts too? The proximity of both the terms themselves and the stages in the life cycle sometimes makes it a bit difficult to understand what is being said, especially as the rest of the paragraph states that : “Another reflection of a rapid development was the fairly few numbers of cells in association or syzygy observed compared to gametocysts”

l. 357-358
“Thus, a trade-off was seen, with rapid events associated with lower production of parasites.”
Could it be that at CW, the parasite phenology is so fast that the gametocysts were already released in the environments within the 1-week window of sampling?

l. 364-366
“ HF gametocytes were larger in size than those at AU (Fig. 5A, HF x = 1016 m3, AU x = 675 um2; Nested ANOVA with parasites nested within earthworms and earthworms nested within sites, P < 0.001).”
There is a problem in the units. Please also add the units in the figure labels fig 5A.

l. 525-540 about Merogony hypothesis.
Although entirely speculative at this stage, the idea of a merogony/schizogony in Monocystis is interesting. There are a few proven cases, indeed in neogregarines but also in Porosporidae (cf Porospora gigantea). There is also the mentionned possibility of vertical transmission. In both cases, I recommend that the authors suggest a number of possible directions for further investigation. In the conclusive parts of the discussion (l. 542-546), I would also suggest that the authors consider/recall the other factors independent of the Monocystis cycle that could explain the observations made (environmental factors, etc.) - or their lack of. This would strengthen the hypotheses presented for discussion.

Additional comments

In summary, this study highlight the inadequacy of existing knowledge of the life cycle and life history of gregarines to account for the data observed in the field. Data on gregarines is scarce, but some cycles, such as that of Monocystis, serve as classic examples. This makes the study all the more interesting, highlighting the need to be cautious with even our most established data, as we know so little about gregarines, even though they are an extremely successful group of parasites!

Reviewer 2 ·

Basic reporting

Submitted manuscript 90979v1 focuses on life the history traits (variations and trade-offs) of gregarines using a model system consisting of the aseptate eugregarine Monocystis perplexa parasitizing the invasive earthworm Amynthas agrestis collected in northern Vermont (USA). Overall, the focus of the manuscript is indeed beneficial, as it fills the gaps in basic knowledge of the gregarine ecology. The manuscript is clearly written, the English used is adequate. However, the manuscript itself requires considerable editing (at least some parts) and cannot be recommended for publication in its current form. Referenced literature used in manuscript is limited to specific topics, further studies should be discussed (some of the studies recommended are listed below or in manuscript PDF). Specific recommendations for improving the manuscript are listed below, others can be found in the edited PDF manuscript.

Experimental design

The experimental design is properly designed and appropriate given the nature of the study.

However, identification methods for both host and parasite species should be provided in Methodology section. While it can be assumed that the identification of the parasite follows the authors’ previous work (Keller & Schall 2020: http://dx.doi.org/10.1645/20-20), the methodology used for parasite identification (molecular and/or microscopic identification) should briefly described with a relevant reference. The same applies to host identification; although there is a subtitle “Host collection and species identification“ (line 199), there is no mention of a method of identifying the host species.

Validity of the findings

While the findings are indeed valuable for the field of gregarine research, the observations and conclusions drawn cannot be generalized for the entire group (Gregarinasina) but only for eugregarines. Neogregarines in particular have a completely different parasitisation strategy (autoinfection can occur here via merogony and autoinfective oocysts), and the infection is often fatal for the host.

RESULTS
Measures of transmission (lines 360-392): While I agree that the size of the gametocyst should correlate with the total number and size of oocysts produced, it does not indicate the viability of those oocysts. This should be taken into account in conclusions. Regarding the trade-off between the number of oocysts produced by the gametocyst and the size of the oocysts (lines 383-385), the total size (volume) and shape of the oocysts can change during their sporulation. Therefore, I wonder, whether authors tested the viability of oocysts (at least in some gametocysts) and their ability to sporulate or even excyst?

DISCUSSION
The authors speculate on reasons why in some cases gregarines stop feeding early, leading to the production of fewer oocysts (lines 476-480). One of the possible answers to this question is the theory of evolution, to which the authors already referred in the Introduction: "Evolutionary theory, for example, predicts that when the host environment deteriorates, the shift to transmission stages would be accelerated (Schneider & Reece, 2021). Accelerated formation of gregarine gametocysts that leave the host body usually occurs due host starving or when the temperature rises above a certain limit. This phenomenon is exploited in preparation of gregarine-free insect colonies (MacDougall 1942: https://doi.org/10.2307/3272778). The size of the parasite cells and the ability to produce a gametocyst depends on the food supply for their host, but also on the nutrition quality (Schreurs & Janovy 2008: http://dx.doi.org/10.1645/GE-1325.1). The results of these and other studies should be considered and discussed in manuscript.

Given the high parasite density in earthworms (lines 518-546), did the authors consider the possibility of autoinfection, as discussed in a recent paper on coelomic eugregarines from polychaete hosts (Valigurová et al. 20232: https://doi.org/10.1016/j.jip.2023.107997)?

CONCLUSIONS (lines 572-573): Development as well as the number of gametocysts produced seem to depend, among other factors, on the response of the host (Jangoux 1987: https://doi.org/10.3354/dao002147; De Ridder & Jangoux 1984: https://doi.org/10.1007/ BF01989307; Coulon & Jangoux 1987: https://doi.org/10.3354/dao002135). Did the authors check for the presence of gregarines/gametocysts in the body cavity of the host, where the gametocysts encapsulated by coelomocytes have often been found in other hosts (including annelids) (e.g. Keilin 1925: https://doi.org/10.1017/S0031182000004510; Valembois et al. 1992: https://doi.org/ 10.1016/0145-305X(92)90010-A; Valigurová et al. 20232: https://doi.org/10.1016/j.jip.2023.107997)?

Additional comments

A table summarizing all abbreviations used in the manuscript would be useful.

Additional comments and requested changes can be found in the edited manuscript PDF file, where parts of the text that require attention are highlighted in yellow.

Annotated reviews are not available for download in order to protect the identity of reviewers who chose to remain anonymous.

---

## Round 0.2 · accepted · Accept

Although one of the previous reviewers was unable to re-review the manuscript, I believe that in your rebuttal you addressed all of that reviewer's comments and the first reviewer also agreed that you adequately addressed all of the comments. The temperature format in the PDF looks good to me.

Reviewer 1 ·

Basic reporting

I think the authors have made very good improvements to their manuscript. The responses and corrections are thorough and relevant. I therefore recommend publication. Congrats to the authors!

Experimental design

no comment

Validity of the findings

no comment